# A Critical Study of the Effect of Polymeric Fibers on the Performance of Supported Liquid Membranes in Sample Microextraction for Metals Analysis

**DOI:** 10.3390/membranes10100275

**Published:** 2020-10-05

**Authors:** Rafael J. González-Álvarez, José A. López-López, Juan J. Pinto, Carlos Moreno

**Affiliations:** Department of Analytical Chemistry, Faculty of Marine and Environmental Sciences, University of Cádiz, República Saharaui s/n, 11510 Puerto Real, Spain; rafael.gonzalezalvarez@uca.es (R.J.G.-A.); joseantonio.lopezlopez@uca.es (J.A.L.-L.); juanjose.pinto@uca.es (J.J.P.)

**Keywords:** hollow fiber supported liquid membrane (HF-SLM), hollow fiber liquid-phase microextraction (HF-LPME), liquid microextraction, metals, preconcentration, silver, speciation

## Abstract

Popularity of hollow fiber-supported liquid membranes (HF-SLM) for liquid-phase microextraction (HF-LPME) has increased in the last decades. In particular, HF-SLM are applied for sample treatment in the determination and speciation of metals. Up to the date, optimization of preconcentration systems has been focused on chemical conditions. However, criteria about fiber selection are not reflected in published works. HFs differ in pore size, porosity, wall thickness, etc., which can affect efficiency and/or selectivity of chemical systems in extraction of metals. In this work, Ag+ transport using tri-isobutylphosphine sulfide (TIBPS) has been used as a model to evaluate differences in metal transport due to the properties of three different fibers. Accurel PP 50/280 fibers, with a higher effective surface and smaller wall thickness, showed the highest efficiency for metal transport. Accurel PP Q3/2 exhibited intermediate efficiency but easier handling and, finally, Accurel PP S6/2 fibers, with a higher wall thickness, offered poorer efficiency but the highest stability and capability for metal speciation. Summarizing, selection of the polymeric support of HF-SLM is a key factor in their applicability of LPME for metals in natural waters.

## 1. Introduction

During the last years, the use of hollow fiber-supported liquid membranes (HF-SLM) in liquid-phase microextraction (LPME) has gained increasing popularity [1,2]. Publications in the field of HF-LPME for trace metals analysis have been increased, and it has become a hot topic due to its efficiency and versatility [1]. In particular, polypropylene (PP) fibers have been widely applied as a support for metal preconcentration, due to the high compatibility of PP with most of solvents, its relatively low cost, and easy handling [3]. HF-LPME devices have been made up in different configurations: solvent bar [4], connected to a syringe [5,6,7], U-shaped [8], and in the form of a loop [9,10,11,12]. The most popular fiber types in trace metals HF-LPME are Accurel PP 50/280 [10,11], Accurel PP Q3/2 [6,9,13], and Accurel PP S6/2 [3] from Membrana^®^, now 3M (Wuppertal, Germany). Other equivalent fiber types have also been used but in less proportion [14].

These fibers have been applied for both purposes: trace metals preconcentration and selective transport of metal species [15]. In particular, fiber Accurel PP Q3/2 and Accurel PP 50/280 have been used mainly for analysis of total concentration of several metals such as Cu, Ni, Cd, Pb, and Ag [9,10,11,13]. In the case of speciation analysis, selective carriers for the free ion have been applied using fiber Accurel PP Q3/2, as it is the case of Hg^2+^ determination using 1-(2-pyridylazo)-2-naphthol as the extractant [16]. Regarding speciation analysis, fiber Accurel PP S6/2 has been used in separation of Ag, Cd, and Pd fractions [4,17,18], as well as for microextraction of Pb, Cr, Pt, Ag, Cd, and Cu.

Normally, optimization of the transport is based on the study of the influence of different physicochemical factors; however, the effect of physical characteristics of the fibers has not been reported, creating a gap of knowledge on how the capillaries used in HF-LPME are selected [6,16,19]. This selection is not trivial, as there are several aspects to be considered regarding the physical properties of the fibers that may affect transport efficiency or even capability for speciation: pore size, wall thickness, effective surface, and porosity [20]. In consequence, having a clear criterion for the selection of the HF offers valuable information for researchers working in the field of HF-SLM applied to metals extraction.

The aim of this work is to offer a critical overview of the effect of fibers characteristics, onto a particular chemical system, applied in HF-LPME. For this reason, a systematic study of the effect of physical properties of fiber Accurel PP 50/280, Accurel PP Q3/2, and Accurel PP S6/2 on silver transport has been carried out. Transport of Ag using TIBPS as a carrier in the organic phase has been previously described and now used as a model system [21,22]. Influence of the different fibers on transport efficiency, speciation capability in the presence of organic matter and salinity, as well as advantages and drawbacks for fibers setup have been discussed. This discussion can be extended to the effect of the fibers on the transport of other metals.

## 2. Materials and Methods

### 2.1. Reagents and Solutions

All reagents were analytical-reagent grade unless otherwise stated. Potassium nitrate (99%) and sodium chloride (99.5%) were obtained from Scharlau (Barcelona, Spain). Kerosene and ammonia (30%) were obtained from Panreac (Barcelona, Spain). Sodium thiosulfate (100%) was purchased from Merck (Darmstadt, Germany). Tri-isobutylphosphine sulfide (TIBPS) was provided by Cytec Industries Inc (Saddle Brook, NJ, USA), and humic acids were obtained from Aldrich (Steinheim, Germany). Aqueous solutions of silver were prepared from a 1000 mg L^−1^ standard solution obtained from Merck (Darmstadt, Germany). Deionized water of resistivity lower than 18.2 MΩ cm was obtained by a Millipore Quantum Ultrapure water supplier (Millipore, Bedford, Ma, USA). Acetylene for atomic spectrometry was obtained from Air Liquide (Madrid, Spain).

### 2.2. Apparatus

Polypropylene Accurel PP 50/280, Accurel PP Q3/2, and Accurel PP S6/2 hollow fibers (Membrana, Wuppertal, Germany) were used as a support for liquid-phase microextraction. Physical properties of the fibers are summarized in Table 1. 

For extraction experiments, samples were stirred in an IKA-Big Squid magnetic stirrer (IKA-Werke, Staufen, Germany). Silver concentration in the aqueous solutions was quantified by flame atomic absorption spectroscopy using a continuum source atomic absorption spectrometer model ContrAA 700 (Analytik Jena, Jena, Germany), using a flame atomizer (FAAS) at a wavelength of 328.1 nm. Humic acids in water samples were quantified by a total organic carbon analyzer Analytik Jena multi N/C 3100.

### 2.3. Set up of the Hollow Fiber-Supported Liquid Membranes Experiments

First of all, and for an appropriate comparison between fiber types, it was decided to recover two different internal volumes from each fiber type (20 and 40 µL). Fiber segments were cut in order to contain the established volume. For all the experiments, a three-phase configuration was used. In this type of configuration, the organic solution containing the carrier is placed into the pores of the fiber, separating the acceptor solution and sample, which are located in the inner and outer parts of the fiber, respectively.

Set up was slightly different depending on the type of fiber used due to their different physical properties and handling. For Accurel PP Q3/2 and Accurel PP S6/2 a three phases solvent bar (3SBME) configuration was used, while a classical HF configuration was used for Accurel PP 50/280.

3SBME fibers were made up as follows [3]:A piece of fiber of the desired length was cut using a blade.One of the fiber’s ends was thermally sealed using a hot tip and tweezers.The fiber was filled with the acceptor solution and the other end was thermally sealed to make-up the 3SBME.The 3SBME was placed into an organic solution containing the carrier for impregnation of the pores.The fiber was left free in the sample for extraction.

In the case of fiber Accurel PP 50/280, both internal diameter and wall thickness physically impede to prepare a 3SBME. Thus, Accurel PP 50/280 fiber was configured as a loop in which the ends of the fiber were sealed using paraffin film to avoid metallic contamination [11]. The process consisted in:The desired fiber length was cut.The lumen of the fiber was filled with the acceptor solution using a microsyringe.In order to impregnate the pores, the fiber was immersed in the organic solution, still connected to the syringe.The lumen was again flushed with acceptor solution and the fiber ends were put together and sealed using paraffin film.The fiber was placed into the sample for extraction.

In order to compare the influence of physical characteristics of hollow fibers on transport efficiency, Ag^+^ transport using TIBPS as extractant was used as a model system [21]. For systematic comparison of the different fiber types, 0.1 mg L^−1^ Ag^+^ samples were prepared by dilution of Ag^+^ from a 1000 mg L^−1^ standard and addition of the corresponding concentration of NO_3_^−^ and humic acids (HA) or NaCl when required. Nitrate was added to the samples because Ag^+^ extraction by TIBPS takes place by cotransport, needing a counter ion for the Ag_2_(TIBPS)_3_^2+^ and Ag(TIBPS)_2_^+^ complexes formed by Ag^+^ and TIBPS [22].

### 2.4. Quantification of Transport Efficiency

Efficiency of silver transport was measured as the enrichment factor (*EF*), which can be defined as the ratio between the concentration of metal inside the fiber after a certain time [*Ag*^+^]*_t_* and the initial concentration of metal in the sample [*Ag*^+^]_0_ (1).
(1)$$EF=\frac{{\left[Ag^{+}\right]}_{t}}{{\left[Ag^{+}\right]}_{0}}\;.$$

Effective surface of organic phase in contact with the sample was calculated taking into account the porosity of the fiber walls (Table 1), as presented in Equation (2), where *S* is the effective surface, *l* is the fiber length, *r* is the internal radius of the fiber, and *p* is the porosity of the fiber wall.
(2)S=2·π·r·l·p.

## 3. Results and Discussion

### 3.1. Influence of Hollow Fiber Characteristics on Transport Efficiency 

In order to assess possible differences due to the fiber characteristics on the performance of the system, the effect of chemical variables and hydrodynamics on the enrichment factor for silver was studied for the three selected fibers. Studied chemical variables were concentration of NO_3_^−^ in the sample, concentration of TIBPS in the organic phase, and concentration of S_2_O_3_^2−^ in the acceptor solution. In general, the profile of *EF* against chemical variables was similar for all the evaluated fibers. As can be observed in Figure 1a, an enhancement in *EF* with NO_3_^−^ concentration up to 0.1 mol L^−1^ was obtained and then it decreased for higher NO_3_^−^ concentration. This has been previously reported for liquid membranes as an effect of the formation of (Ag_2_(TIBPS)_3_(NO_3_)_2_) at NO_3_^−^ concentration higher than 0.1 mol L^−1^, which presents lower permeability through the organic membrane than (Ag(TIBPS)_2_NO_3_) [22]. In the case of TIBPS, the extraction of Ag^+^ increased with the concentration of the extractant up to 0.1 mol L^−1^ TIBPS (Figure 1b) and then, no significant variation was observed. This behavior may be related to the lower viscosity of TIBPS at low concentrations. Finally, when the concentration of S_2_O_3_^2−^ was varied in the acceptor solution, *EF* presented a maximum at 0.1 mol L^−1^ and decreased at higher concentration, probably due to the formation of nonextractable silver complexes (Figure 1c).

In all cases, the highest *EF* was obtained using fiber Accurel PP 50/280 (Figure 1). This is supported by a smaller wall thickness and a higher effective surface. Wall thickness is one of the limiting factors of hollow fiber liquid-phase microextraction, because diffusion through the organic phase immobilized in the fiber pores cannot be mechanically enhanced [3]. Additionally, higher effective surface resulted in a better contact between the transported species at both interfaces. According to this observation, Accurel PP Q3/2 offered an intermediate *EF*, while the poorest preconcentration was obtained for Accurel PP S6/2.

Regarding hydrodynamics, the effect of stirring speed in the sample on the *EF* was evaluated. Figure 2a shows an enhancement of *EF* up to 800 rpm and then it decreased due to the removal of the organic phase from the fiber pores. Evaluating the effect of time on *EF* (Figure 2b), maximum preconcentration was obtained at 4 h of extraction, showing a decrease in the *EF* for longer times. This has been previously observed for HF-LPME systems due to a loss of the organic phase from the fiber pores for long-time experiments [11,23]. It is worth to mention that in the case of Accurel PP S6/2, the enrichment factor experienced a smaller relative decrease at high stirring speed than that for Accurel PP Q3/2 (both with pore size 0.2 µm), suggesting that the higher wall thickness allows a higher stability of the organic solution into the fiber pores, which also has been observed in the literature [17,18].

Thus, fibers could be ordered on the basis of efficiency in the transport of Ag^+^ as: Accurel PP 50/280 > Accurel PP Q3/2 > Accurel PP S6/2. Despite their potential, Accurel PP 50/280 have been less used in environmental analysis of metals than Accurel PP Q3/2 because of difficulties in the set up [10].

In this work, 20 and 40 µL acceptor solutions containing fibers Accurel PP 50/280, Accurel PP Q3/2, and Accurel PP S6/2 were also prepared. Initially, it could be thought that increasing the volume ratio between the sample and acceptor solution could result in higher *EF*. However, fibers containing 40 µL acceptor solution offered the highest *EF*. This could be explained by the prevalence of the effect of higher effective surface over the dilution effect.

### 3.2. Relationship between the Effect of Organic Matter Concentration and Polymeric Support

An important aspect to be taken into account in the application of polypropylene hollow fibers for Ag^+^ analysis in natural waters is associated with the pore size of the fiber walls. Pores size of Accurel PP Q3/2 and Accurel PP S6/2 is 0.2 µm, while Accurel PP 50/280 pores are 0.1 µm. Thus, it could be thought that size exclusion could take place in the cases that silver is forming macromolecular or colloidal complexes with humic acids. Experiments were carried out using samples containing organic matter in the form of humic acids, measured as dissolved organic carbon (DOC) within the range from 0.3 to 40 mg L^−1^ of DOC. In all cases, time of experiment was 120 min.

Fiber Accurel PP 50/280 presented the highest *EF*, followed by Accurel PP Q3/2 and Accurel PP S6/2 as it was previously observed in the absence of HA (Figure 3). This means that differences of effective surface between the fibers are of main importance for keeping their capability in the transport of the metal. Similar evolution of *EF* with increasing HA concentration was observed for all fibers studied, being particularly important in the range of 0.3–1.5 mg L^−1^ DOC, where organic complexes appear and reach higher concentration. From the results obtained, the observed decrease in *EF* with increasing HA concentration cannot be related with different pore size of the studied fibers. Therefore, the effect of HA on *EF* could be explained by the formation of Ag-HA complexes, which are not transported through the organic phase due to a low solubility, leading to a selective transport of the free ion from the sample [4,23,24].

### 3.3. Relationships between the Effect of Sample Salinity and Polymeric Support

Salinity is a main factor affecting the transport of metal cations by liquid membranes and liquid-phase microextraction in natural waters and, in particular, in marine samples [24,25]. When this effect has been quantified, it may be used to evaluate the extent of metal complexation in saline systems such as estuaries and seawater [17]. In the case of silver, hollow fiber liquid-phase microextraction using fiber Accurel PP S6/2 allowed selective transport of Ag^+^ in the presence of AgCl*_n_*^(*n*−1)−^ that can be formed due to increasing salinity in estuaries in the direction to the river mouth [7,21].

Figure 4 shows that the relative order of efficiency for silver transport is Accurel PP 50/280 > Accurel PP Q3/2 > Accurel PP S6/2 mainly due to higher effective surface for the first fiber type. However, a different trend of *EF* with Cl^−^ concentration was observed within the fibers. On the one hand, for Accurel PP S6/2, a decrease in *EF* was obtained with increasing Cl^−^ concentration. On the other hand, for fibers Accurel PP Q3/2 and Accurel PP 50/280, an enhancement of *EF* with Cl^−^ concentration was observed. Reasons for this difference are not clear, and further research is needed to get deeper knowledge about how differences between fibers lead to it. Notwithstanding, possible causes could be related with differences in wall thickness.

The increase in *EF* with Cl^−^ concentration obtained for Accurel PP 50/280 and Accurel PP Q3/2 could be associated with the formation of complexes as AgCl*_n_*^(*n*−1)−^ that could be potentially transported through the membrane as AgCl*_n_*−TIBPS species additionally to Ag^+^. However, wall thickness in the case of Accurel PP S6/2 could be high enough to allow a kinetic separation of Ag-TIBPS from AgCl*_n_*^(*n*−1)−^—TIBPS [22]. This fact reveals the importance of a proper selection of the fiber type to be used in preconcentration studies with analytical purposes.

Finally, with an increase in Cl^−^ concentration, the highest *EF* was obtained for fibers Accurel PP 50/280 and Accurel PP Q3/2 containing 20 µL acceptor solution. In this case, the volume ratio between the sample and the acceptor solution prevailed on the effect of effective surface. This could be explained by the additional transport of AgCl*_n_*^(*n*−1)−^ species through the organic phase formed due to the presence of Cl^−^ in the sample.

### 3.4. Selection of Fiber Type

Selection of the fiber type should be carried out on the basis of their fit for analytical purpose and the instrumental analysis facilities available. In this case, we first fixed the volume of receiving solution (internal volume) for each fiber and then fiber length was adjusted as required. For Accurel PP S6/2, 1–2 cm fibers were used meaning that special care has to be paid during fibers set up to ensure reproducibility.

As far as it has been published, there are no clear criteria for selection of a hollow fiber type in the preconcentration of metals. Notwithstanding, Accurel PP Q3/2 have been the most widely used fiber [5,6,7,8,9], probably be due to its easier handling if compared with Accurel PP 50/280 and its smaller internal diameter if compared with Accurel PP S6/2. As a consequence, they are expected to offer better balance between reproducibility and *EF*. Advantages of Accurel PP 50/280 over the other Accurel fiber types are based on a higher effective surface as it has been proved for silver [11]. As an example, this type of fiber has allowed obtaining preconcentration factors of 1000 times for Ag in fresh waters and 260 times in the case of Pb [10,11]. Regarding Accurel PP S6/2, it offers an easier handling and stability that has led to its application for 3SBME systems, as well as speciation capability [17,18,21,22,23,24]. In addition, this configuration allows the performance of higher number of experiments without requiring the use of a support for fibers.

## 4. Conclusions

Selection of the proper fiber type is of main concern due to their different physical characteristics. Differences may affect important factors of analytical methods, such as preconcentration efficiency, applicability, suitability for speciation, and reproducibility.

Summarizing Accurel PP Q3/2 fibers offer good performance for set up, stability, and preconcentration capability, whereas Accurel PP 50/280 offer the best efficiency due to a higher effective surface but a more difficult handling. Finally, Accurel PP S6/2 fibers present less potential for preconcentration, but higher capacity to separate metallic species. Additionally, this fiber is more adaptable to different configurations and presents a better performance for automation.

## Figures and Tables

**Figure 1 membranes-10-00275-f001:**
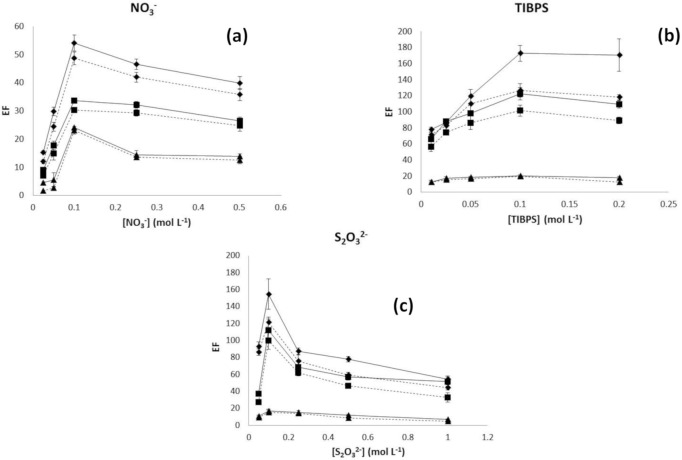
Variation of enrichment factor (*EF*) with (**a**) NO_3_^−^ for tri-isobutylphosphine sulfide (TIBPS) 0.01 M, S_2_O_3_^2−^ 0.1 M, (**b**) TIBPS for NO_3_^−^ 0.1 M, S_2_O_3_^2−^ 0.1 M, and (**c**) S_2_O_3_^2−^ for NO_3_^−^ 0.1 M, TIBPS 0.1 M for tested hollow fibers: ♦ Accurel PP 50/280, ■ Accurel PP Q3/2 ▲ Accurel PP S6/2. **Dotted line** represents 20 μL acceptor solution and **S****olid line** represents 40 μL acceptor solution.

**Figure 2 membranes-10-00275-f002:**
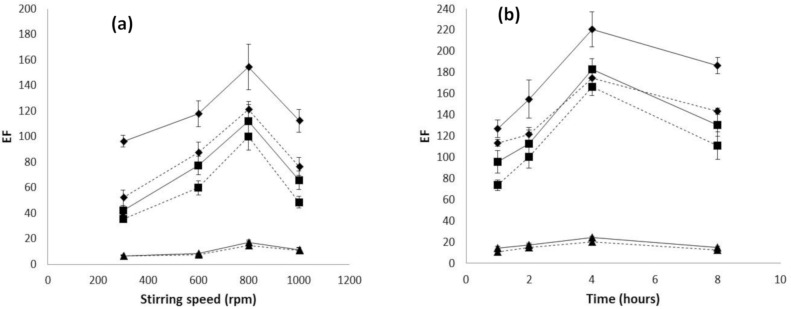
Variation of *EF* with (**a**) stirring speed and (**b**) time for the three evaluated hollow fiber types: ♦ Accurel PP 50/280, ■ Accurel PP Q3/2, and ▲ Accurel PP S6/2. **Dotted line** represents 20 μL acceptor solution and **S****olid line** represents 40 μL acceptor solution.

**Figure 3 membranes-10-00275-f003:**
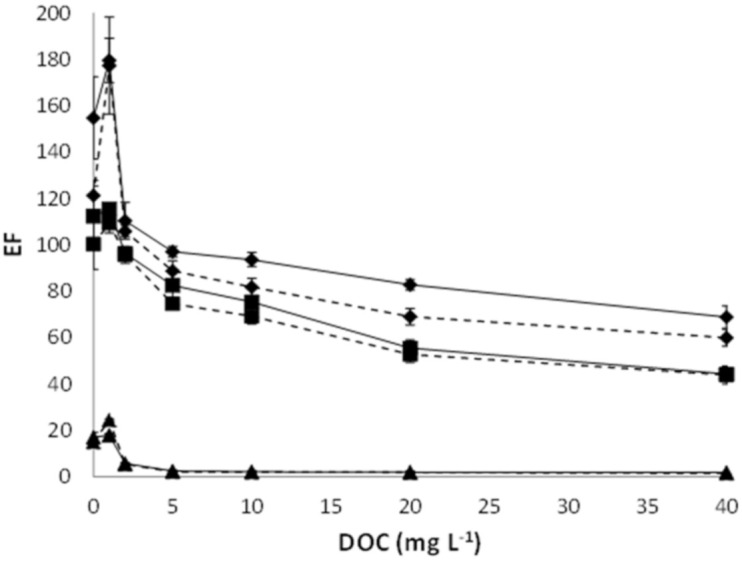
Variation of *EF* against DOC concentration in samples for the three evaluated hollow fiber types: ♦ Accurel PP 50/280, ■ Accurel PP Q3/2, and ▲ Accurel PP S6/2. Time of extraction 120 min. **Dotted line** represents 20 μL acceptor solution and **S****olid line** represents 40 μL acceptor solution

**Figure 4 membranes-10-00275-f004:**
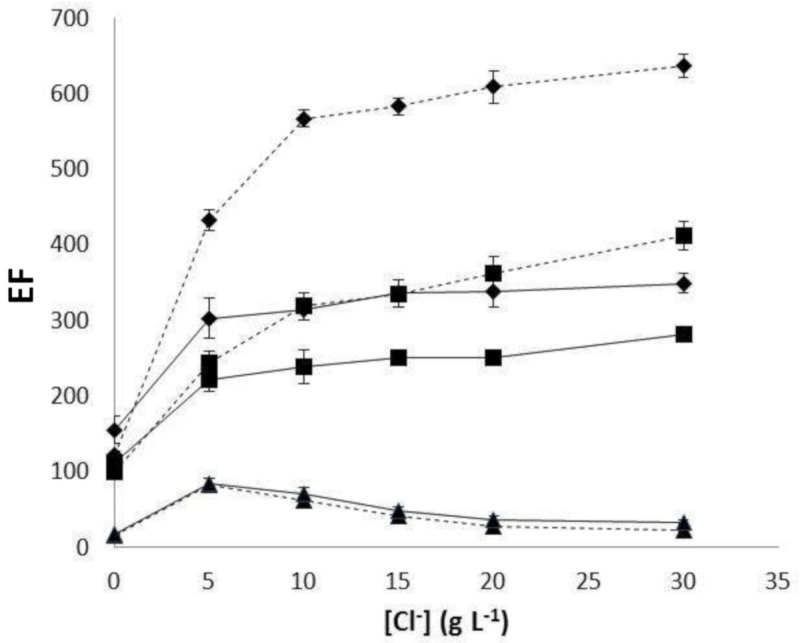
Variation of *EF* against Cl^−^ concentration in samples for the three evaluated hollow fiber types: ♦ Accurel PP 50/280, ■ Accurel PP Q3/2, and ▲ Accurel PP S6/2. Time of extraction 120 min. **Dotted line** represents 20 μL acceptor solution and **S****olid line** represents 40 μL acceptor solution.

**Table 1 membranes-10-00275-t001:** Physical characteristics of the hollow fibers used in this study.

Fiber	Internal Diameter (µm)	Pore Size (µm)	Porosity (%)	Wall Thickness (µm)	Effective Surface (mm^2^)	Length (mm)
50/280	280	0.1	60	50	211	400
Q3/2	600	0.2	75	200	127	90
S6/2	1800	0.2	72	450	41	10

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
