# Peer review of "A Critical Study of the Effect of Polymeric Fibers on the Performance of Supported Liquid Membranes in Sample Microextraction for Metals Analysis"

_membranes, 2020, doi:10.3390/membranes10100275_

Round 1

Reviewer 1 Report

In this work the effect of polymeric fibers on the performance of supported liquid membranes in sample micro-extraction for is studied using Ag+ transport with tri-isobutylphosphine sulfide (TIBPS) as model system. Three hollow fibers with different properties were employed to decide which of them is better for metal analysis. Influence of the different fibers on transport efficiency, speciation capability in the presence of organic matter and salinity, as well as advantages and drawbacks for fibers set up have been discussed. It was concluded that the selection of the polymeric support is a key factor in their applicability of LPME for metals in natural waters. As the work is well performed, clearly written, and contributes in the application field of metal analysis, I recommend its publication in Membranes.

Some minor points to address to increase the overall rate of the manuscript are:

  1. In lines 144-148 an explanation for the profiles of the effect of TIBPS and S2O3 2- is missing.
  2. Along different figures (1-4) the black circle symbol of the Accurel PP 50/280 lines in the legends should be a black diamont to correspond with the figures.
  3. In line 169 it is stated that “a smaller relative decrease at high stirring speed than for the other fibers” is observed for the Accurel PP S6/2 fiber. However, I think that is relative terms (percents ) this decrese may be the same for all fibers, as the Acucurel PP S6/2 has the lowest EF values.
  4. In lines 183-186, it is stated that “Initially, it could be thought that increasing volume ratio between the sample and acceptor solution could result in higher EF”. There are several effects to consider: a) as the volume of the acceptor solution increases an increse in back-extraction of the metal from the organic phase is expected as the phase ratio (organic volume/acceptor volume) decreses according to SX theory; b) as the acceptor volume increses there is a dilution of metal concentration in this phase; however, if permeability is constant the percent of metal in the acceptor phase is the same, c) as authors said, as the volume increseses the interfacial area increases, but a constant permeability assures no effect of the permeation area. Could the authors give a relationship (equation) to relate EF with permeability to try to understand the effect of the different factors involved?
  5. In Figure 3, could the authors give an explanation for the strong decrese in EF with DOM in the range 0.3-1.5 mg L-1 and then it almost constant value?
  6. In section 3.3 the effect of salinity is studied. Different trend were observed for the different fibers which were not clear. Although a possible explanation based on wall thickness is indicated based on kinetics effects is provided, are the studied systems under a kinetic regime, a diffusion or a mixed one? Is there a possible way to verify this to confirm authors’ hyphothesis?
  7. In line 145, the work “the” is probably “then”.

Author Response

The authors acknowledge reviewer's comments, which will allow a better presentation of our results.

Corrections made to the text have been highlighted in red

COMMENTS

- In lines 144-148 an explanation for the profiles of the effect of TIBPS and S2O3 2- is missing.

The text has been revised and improved as suggested.

- Along different figures (1-4) the black circle symbol of the Accurel PP 50/280 lines in the legends should be a black diamont to correspond with the figures.

The symbol has been corrected in the legends of figures 1-4.

- In line 169 it is stated that “a smaller relative decrease at high stirring speed than for the other fibers” is observed for the Accurel PP S6/2 fiber. However, I think that is relative terms (percents ) this decrese may be the same for all fibers, as the Accurel PP S6/2 has the lowest EF values.

Thanks to reviewer’s comment, since we were able to find a mistake in the manuscript. The effect described was not in general, but a comparison between S6/2 fiber and Q3/2. The text has been now corrected.

- In lines 183-186, it is stated that “Initially, it could be thought that increasing volume ratio between the sample and acceptor solution could result in higher EF”. There are several effects to consider: a) as the volume of the acceptor solution increases an increse in back-extraction of the metal from the organic phase is expected as the phase ratio (organic volume/acceptor volume) decreses according to SX theory; b) as the acceptor volume increses there is a dilution of metal concentration in this phase; however, if permeability is constant the percent of metal in the acceptor phase is the same, c) as authors said, as the volume increseses the interfacial area increases, but a constant permeability assures no effect of the permeation area. Could the authors give a relationship (equation) to relate EF with permeability to try to understand the effect of the different factors involved?

Although reviewer’s suggestion seems to be quite interesting, in this work we have explained the effect mentioned in a qualitative way. To establish a quantitative relationship (equation) with the available data of the experiments could be in some way speculative, since that was not our aim.

- In Figure 3, could the authors give an explanation for the strong decrese in EF with DOM in the range 0.3-1.5 mg L-1 and then it almost constant value?

0.3 ppm was the first concentration of the studied range and then, the strong decrease was related with the absence/presence of organic complexes. This has been better explained in the text.

- In section 3.3 the effect of salinity is studied. Different trend were observed for the different fibers which were not clear. Although a possible explanation based on wall thickness is indicated based on kinetics effects is provided, are the studied systems under a kinetic regime, a diffusion or a mixed one? Is there a possible way to verify this to confirm authors’ hyphothesis?

We acknowledge reviewer’s comment, since this could be an interesting subject to studied. As mentioned in the text, we think that additional research is need to clarify this effect, and may be a new and specific work. Thus, in the present work we have addressed the practical aspect of the selection of the fibers.  

- In line 145, the work “the” is probably “then”.

The error has been corrected.

Reviewer 2 Report

This work investigated the effect of polymeric fibers on the performance of supported liquid membranes in sample micro-extraction for metals analysis. The effect of physical properties of fibers Accurel PP 50/280, Accurel PP Q3/2, and Accurel PP S6/2 on silver transport has been carried out. This work is suggested to be considered to publish in the journal with careful revisions. Following are some comments:

  1. The experiment conditions in Figure 1 should be listed in detail. Why the data in Figures 1(a), 1(b), and 1(c) seems have no connection. For example, the data of PP S6/2 in the three Figures were entirely different. The effect of the factors should be optimized one by one.
  2. The icon for PP 50/280 in all Figures is inconsistent with their Figure caption.
  3. The data in Figure 3 should start from zero, because the authors expounded in the lines 195-196 of page 5 that “… previously observed in the absence of HA (Figure 3).”
  4. It is inconsistent that the use of humic acids. Since the humic acids have complexation towards Ag+ ions, it cannot distinguish the decrease of EF with the HA concentration due to their complexation or the different pore size. It is easy to know that even the micromolecule complexing agent EDTA can also decrease the EF, which is not related to the different pore size.
  5. The extraction and back-extraction reaction equations should be given.
  6. The possible explanation of the effect of Cl- concentration for PP S6/2 should be given.

Author Response

The authors acknowledge reviewer's comments, which will allow a better presentation of our results.

Corrections made to the text have been highlighted in red

COMMENTS

- The experiment conditions in Figure 1 should be listed in detail. Why the data in Figures 1(a), 1(b), and 1(c) seems have no connection. For example, the data of PP S6/2 in the three Figures were entirely different. The effect of the factors should be optimized one by one.

Some experimental conditions were missed and now have been included, giving coherence to the represented data.

- The icon for PP 50/280 in all Figures is inconsistent with their Figure caption.

The error has been corrected.

- The data in Figure 3 should start from zero, because the authors expounded in the lines 195-196 of page 5 that “… previously observed in the absence of HA (Figure 3).”

Figure has been changed as suggested.

- It is inconsistent that the use of humic acids. Since the humic acids have complexation towards Ag+ ions, it cannot distinguish the decrease of EF with the HA concentration due to their complexation or the different pore size. It is easy to know that even the micromolecule complexing agent EDTA can also decrease the EF, which is not related to the different pore size.

Please note that metal complexation in general do not decrease EF. This will happened only if complexes are not transported through the organic membrane (charge, size, etc.)   

- The extraction and back-extraction reaction equations should be given.

Since the system was previously described, we have added a new reference (line 57).

- The possible explanation of the effect of Cl- concentration for PP S6/2 should be given.

As mentioned in the text, we think that additional research is need to clarify this effect, and may be a new and specific work. In the present work we have addressed the practical aspect of the selection of the fibers and suggested a possible explanation of the effect.

Reviewer 3 Report

membranes-948905

This study is a natural follow-up for a lead research group in this area, the need for a study of suitability of different types of membrane fibers for metal ions is needed, and the work seems valid. Therefore this is a suitable manuscript for publication. However, the English grammar needs major revision, and must be corrected before acceptance. The authors may want to send it out to a native-born English speaking scientist for final review before re-submission. Some of the suggested corrections are listed below. These are not all of them. There are also a few questions regarding missing material.

  1. Hollow fiber liquid phase microextraction is, by convention, now abbreviated HF-LPME. Please correct throughout the manuscript.
  2. The article “the is incorrectly used in several places in the manuscript: lines 15, 16,18 as examples, or missing: line 183.
  3. The word fiber(s) is out of order: lines 39, 42, 43, 55, 104, 155, 195, 216. 222
  4. There are a number of examples of poor word or phrase choices: line 247-249 for example.

Questions:

  1. Was a metal needle used for the syringe: line 109.
  2. To be accurate, include what the fiber is being used for on line 283.
  3. The keywords should include hollow fiber liquid phase microextraction (HF-LPME).
  4. Despite these authors being leading researchers in this venue, it is a little surprising that they chose to use 8 of 25 references from their own publications.

Author Response

The authors acknowledge reviewer's comments, which will allow a better presentation of our results.

Corrections made to the text have been highlighted in red

COMMENTS

- Hollow fiber liquid phase microextraction is, by convention, now abbreviated HF-LPME. Please correct throughout the manuscript.

Text has been corrected as suggested.

- The article “the is incorrectly used in several places in the manuscript: lines 15, 16,18 as examples, or missing: line 183.

Text was revised and corrected as suggested.

- The word fiber(s) is out of order: lines 39, 42, 43, 55, 104, 155, 195, 216. 222

Text was revised and corrected as suggested.

- There are a number of examples of poor word or phrase choices: line 247-249 for example.

Text was revised and corrected as suggested.

- Was a metal needle used for the syringe: line 109.

An insulin-type microneedle was used. From previous studies, no influence of the needle in silver concentrations was observed.

- To be accurate, include what the fiber is being used for on line 283.

We did not understand reviewer’s comment, since line 283 corresponds to a reference.

- The keywords should include hollow fiber liquid phase microextraction (HF-LPME).

New keyword has been included

- Despite these authors being leading researchers in this venue, it is a little surprising that they chose to use 8 of 25 references from their own publications.

As reviewer said, our research group has worked extensively in this subject. To prepare the manuscript, we have selected the required bibliography among the references available, and without taking into account affiliations.

Reviewer 4 Report

I think this paper is interesting and can be published after the following minor revision:

1) Line 30, please correct "have been increased has becom"

2) Line 33, please correct "HLPME" to "HFLPME"

3) Line 39, please clarify what is "transport of metal fractions" as this is not clear

4) Line 67, please add city where Cytec is located

5) Line 70, please add city and state where Millipore is located

6) Line 81, please add city where Ika is located

7) Line 83, please add city where Jena Analytik is located

8) Line 85, please remove "(Analytik Jena, Germany)"

9) Line 96, please correct "ups"

10) Line 212-13, please correct "liquid micro-extraction" to "liquid-phase micro-extraction"

Author Response

The authors acknowledge reviewer's comments, which will allow a better presentation of our results.

Corrections made to the text have been highlighted in red

COMMENTS

- Line 30, please correct "have been increased has becom"

Text has been revised as suggested

- Line 33, please correct "HLPME" to "HFLPME"

Text has been revised as suggested

- Line 39, please clarify what is "transport of metal fractions" as this is not clear

Metal fractions is also used to refer chemical species, specially to differentiate organic and ionorganic or free species. For a better understanding, we has used “metal species”

- Line 67, please add city where Cytec is located

Text has been revised as suggested

- Line 70, please add city and state where Millipore is located

Text has been revised as suggested

- Line 81, please add city where Ika is located

Text has been revised as suggested

- Line 83, please add city where Jena Analytik is located

Text has been revised as suggested

- Line 85, please remove "(Analytik Jena, Germany)"

Text has been revised as suggested

- Line 96, please correct "ups"

Error has been corrected

- Line 212-13, please correct "liquid micro-extraction" to "liquid-phase micro-extraction"

Text has been revised as suggested